# Associations between Ethnicity and Referrals, Access and Engagement in a UK Adult Burns Clinical Psychology Service

**Laura Shepherd** [1,*] , **Ishani Hari** [2] **and Lauren Bamford** [3]

1   Department of Clinical Psychology & Neuropsychology, Queens Medical Centre,
    Nottingham University Hospitals NHS Trust, Derby Road, Nottingham NG7 2UH, UK
2   Academic Neuroscience Department, Queens Medical Centre, Nottingham University Hospitals NHS Trust,
    Derby Road, Nottingham NG7 2UH, UK
3   Department of Neuroscience, Psychology and Behaviour, George Davies Centre, University of Leicester,
    Lancaster Road, Leicester LE1 7HA, UK
*   Correspondence: laura.shepherd@nuh.nhs.uk

**Abstract:** Ethnic inequalities exist across healthcare, including access to and experiences and outcomes of mental health services. Access to and engagement with burns clinical psychology services is essential for all patients. This study aimed to explore the ethnic diversity of adults referred to a burns clinical psychology service compared to those admitted to the burns service. It also aimed to investigate associations between ethnicity and indicators of access and engagement (receiving, declining or not attending psychological assessments, receiving psychological therapy and the number of therapy sessions completed). Routinely collected data over eight years were analysed. Analysis revealed an association between ethnicity and referral to the burns clinical psychology service. Patients from White British and Other ethnic backgrounds were less likely to be referred, whereas patients from Black and Asian ethnic groups were more likely to be referred. There were no statistically significant associations between ethnicity and receiving, declining or not attending psychological assessments or receiving psychological therapy. Furthermore, there was no statistically significant difference in the number of psychological therapy sessions received between the ethnic groups. Therefore, patients from ethnic minority groups did not appear to have significant difficulties engaging with the service but further research is recommended.

**Keywords:** ethnicity; ethnic group; psychology; psychological; referral; access; engagement



## 1. Introduction

Ethnicity is defined as a group of people who share common national or cultural traditions [1], and ethnic diversity is the identified differences between individuals from various ethnic groups [2]. Ethnic inequalities are known to exist across healthcare within the United Kingdom (UK) and this includes access to, experiences of, and outcomes of psychological therapy or mental health services [3]. Reasons posited include barriers to access, lack of interpreting services and a distrust of services [4]. Studies have also shown that some ethnic minority groups delay seeking help from mental health services due to the perception that healthcare professionals do not understand racism, that racist experiences impact engagement with services and a sense of frustration with having to explain ethnicity-related experiences repeatedly [5,6]. Additionally, ethnic minority groups have been reported to be less likely to refer themselves or be referred to services by their general practitioner compared to patients from a White British background [3].

This discrepancy has been shown to extend to aspects of service delivery. For example, patients from ethnic minority groups with psychosis have been found to be offered psychological therapy less frequently compared to White British patients [7]. Evidence also

suggests that inequalities in mental health services continue across the lifespan, such as the similar patterns of discrimination experienced by Black men from adolescence and into adulthood [8,9]. Furthermore, repeated episodes of racism and racial discrimination have been associated with poorer physical and mental health outcomes [10,11]. It is therefore vital for all healthcare services to be aware of and explore any potential ethnic inequalities, including mental health and other services involving psychological support.

Considering burn injuries, there are a number of studies that have explored ethnicity in relation to the risk of sustaining a burn injury and burn injury outcomes. An early study identified that being from an ethnic minority group was associated with an increased risk of sustaining a burn injury, alongside other socioeconomic factors such as low income, a large family structure, unemployment and substandard living conditions [12]. Another study, based on 46 burn centres in North America, found that White patients were the largest group, had the largest proportion of flame burns and shared the highest mortality rate with indigenous patients, whereas patients from indigenous backgrounds had the highest rates of full thickness burns and had longer hospital and intensive care stays [13]. Another study using the same data registry reported that African-American race was associated with high mortality rates, along with socioeconomic status and gender [14].

Studies conducted in the UK have also reported similar findings. For example, a study on burn injury admissions in London revealed that patients from ethnic minority groups had the highest rate of burn injury, with socioeconomic deprivation and geographical location also increasing risk [15]. In a systematic review of studies on burn epidemiology in the UK and high-income countries within North America, Australia and Europe, it was reported that children from ethnic minority backgrounds often had more severe burns or longer hospital stays [16]. Furthermore, a review of paediatric patients attending a burns service in another area of the UK found that children from ethnic minority groups had higher total body surface area burns and an increased length of stay compared to non-ethnic minority children [17].

In relation to disparities in outcomes following burn injuries, a recent systematic review of the literature exploring ethnicity and burn injury outcomes concluded that most studies showed poorer inpatient and outpatient outcomes in burns patients from ethnic minority backgrounds [18]. Furthermore, a large study conducted in North America suggested that racial and ethnic minority status was associated with lower satisfaction with appearance and reduced social community integration compared to White patients at baseline and 6, 12 and 24 months post burn injury [19]. Additionally, it is reported that Black and Hispanic patients may need particular support in community reintegration following burn injuries as findings revealed that White patients had higher community integration over time compared to these ethnic minority groups [20].

Psychological care is considered an essential component of burns care, reflected in guidance within the UK [21], Europe [22] and worldwide [23,24]. This is due to research consistently showing that patients who have experienced burn injuries can experience persistent psychological and mental health difficulties, including anxiety, depression, post-traumatic stress disorder symptoms and distress associated with pain or scarring [25,26]. Therefore, understanding and addressing potential barriers to accessing and engaging in burns clinical psychology services is imperative, especially considering research showing that patients from ethnic minority groups report barriers accessing and engaging with mental health services more generally [3–11].

Given the widespread understanding that psychological support is essential for the burns population, that burn injuries and burn injury outcomes can be associated with ethnicity, and that patients from ethnic minority groups generally have difficulties accessing and engaging with psychological and mental health services, this study aimed to explore the role of ethnicity in referrals to and engagement with a burns clinical psychology service in the UK over an eight-year period. No previous published research on this topic could be found. Specifically, this study aimed to explore associations or differences between ethnicity and indicators of access and engagement with the service.

## 2. Materials and Methods

### 2.1. Data Extraction

The project was registered as a clinical audit by Nottingham University Hospitals NHS Trust in the United Kingdom (ref: 22-332C). A retrospective analysis of routinely collected clinical data within the burns service in Nottingham, England, over an eight-year period (May 2014 to May 2022) was conducted. Two clinical databases were used to extract relevant data for analysis.

Data were extracted from a database capturing adult burns clinical psychology referrals over the audit period, including gender, age, ethnicity, whether patients attended, declined or did not attend a psychological assessment, whether patients received any psychological therapy sessions and how many psychological therapy sessions were completed. Ethnicity data were also extracted from a second clinical database documenting all adult patients admitted to the affiliated burns service over the same time period. This served as a comparator to patients referred to the burns clinical psychology service.

It is acknowledged that language and terminology are important when discussing ethnicity. The ethnicity data used within this study had been initially categorised according to descriptors used by Nottingham University Hospitals NHS Trust. For the purposes of the study, patients were further categorised into one of seven groups: White British; Other White; Black ethnic group; Asian ethic group; Mixed ethnic origin; Other ethnic group; and Unknown. These categories of ethnic groups are suggested by the UK government [27].

### 2.2. Inclusion Criteria

All patients referred to the adult burns clinical psychology service and admitted to the affiliated burns service over the audit period were included. Within the affiliated adult burns service, patients are typically 18 years or over. However, patients aged 16 or 17 years old may also be admitted if an adult ward is deemed more appropriate than a paediatric ward.

### 2.3. Data Analysis

Frequencies and descriptive statistics were used to explore how both patients admitted to the affiliated burns service and patients referred to the burns clinical psychology service were split in relation to gender, age and ethnicity. Frequencies and percentages of patients were used to represent who attended, declined or did not attend psychological assessments and received psychological therapy after being referred to the burns clinical psychology service across six different ethnic groups (excluding the Unknown group).

To explore associations and differences between ethnicity and indicators of access and engagement, patients whose ethnicity was Unknown were excluded and analyses were completed on the remaining six groups of patients.

Statistical Package for Social Sciences (SPSS) version 26 was used to analyse the data. Pearson's chi-square, Fisher's exact and Kruskal–Wallis analyses were used to explore associations and differences between the groups. Chi-square analyses were used when at least 80% of expected cell values were five or higher. A Fisher's exact test was used when this assumption was not met. Chi-square post hoc analyses were completed using the multiple comparison method suggested by Beasley and Schumacker [28]. Statistical significance was set at $p \leq 0.05$, apart from when post hoc analyses were used, in which case Bonferroni corrections were made to account for multiple testing and reduce the chances of obtaining false-positive results (type I errors).

## 3. Results

### 3.1. Referrals to the Burns Clinical Psychology Service by Ethnic Group

Over the eight-year audit period, 699 adult patients were referred to the burns clinical psychology service. Overall, 55% ($n = 383$) were male, 45% ($n = 315$) were female and one patient's gender was unstated. The average age of patients was 43 years (range: 16–97 years). Over the same period, there had been 1731 patients admitted to the affiliated burns service.

Among patients, 65% (*n* = 1122) were male, 35% (*n* = 608) were female and one patient's gender was unstated. The average age of patients was 46 years (range: 16–102 years).

Table 1 presents information related to the ethnic diversity of patients referred to the burns clinical psychology service and admitted to the affiliated burns service.

**Table 1.** Ethnicity of patients referred to the burns clinical psychology service and admitted to the affiliated burns service.

| Ethnic Group | No. of Burns Clinical Psychology Referrals (%) | No. of Burns Service Admissions (%) |
|---|---|---|
| White British | 471 (67.4%) | 1135 (65.6%) |
| Other White | 38 (5.4%) | 64 (3.7%) |
| Black ethnic group | 28 (4.0%) | 33 (1.9%) |
| Asian ethnic group | 43 (6.2%) | 57 (3.3%) |
| Mixed ethnic origin | 10 (1.4%) | 13 (0.8%) |
| Other ethnic group | 9 (1.3%) | 55 (3.2%) |
| Unknown | 100 (14.3%) | 374 (21.6%) |
| Total | 699 (100%) | 1731 (100%) |

Table 1 shows that the proportion of those referred to the burns clinical psychology service whose ethnicity was White British was 67.4%, whereas the proportion of White British patients admitted to the affiliated burns service was 65.6%. The proportion of patients referred to the burns clinical psychology service from ethnic minority groups (Other White, Black ethnic group, Asian ethnic group, Mixed ethnic origin and Other ethnic group) was 18.3%, whereas the proportion of patients from ethnic minority groups admitted to the affiliated burns service was 12.9%.

Table 1 also highlights how ethnicity was unknown for a significant proportion of patients. For those referred to the burns clinical psychology service, ethnicity was unknown in 14.3% of patients, whereas for patients admitted to the affiliated burns service this was 21.6%.

Pearson's chi-square analysis, excluding patients in the Unknown ethnicity group to avoid confounding the results, revealed a statistically significant association between ethnicity and being referred to the burns clinical psychology service ($\chi^2$ (5, *n* = 1357) = 76.94, *p* = < 0.001). Post hoc analyses were conducted and the p value required for statistical significance was set at *p* = 0.0042 to account for multiple testing. These analyses revealed that patients within the White British ($\chi^2$ = 19.62, *p* < 0.001) and Other ($\chi^2$ = 17.98, *p* = < 0.001) ethnic groups were less likely to be referred to the burns clinical psychology service, whereas patients within Black ($\chi^2$ = 22.75, *p* = < 0.001) and Asian ($\chi^2$ = 23.62, *p* = < 0.001) ethnic groups were more likely to be referred. There was no statistically significant association between referral and being of Other White ethnicity ($\chi^2$ = 6.30, *p* = 0.012) or Mixed ethnic origin ($\chi^2$ = 5.71, *p* = 0.017).

*3.2. Access and Engagement with the Burns Clinical Psychology Service*

Relationships between ethnicity and indicators of access and engagement with the burns clinical psychology service were explored. Patients whose ethnicity was Unknown were again excluded. Indicators of access and engagement included receiving, declining and not attending psychological assessments, receiving psychological therapy and the number of psychological therapy sessions completed.

Table 2 displays these indicators across the six different ethnic groups.

Table 2 suggests that some groups besides the White British group were below the mean in terms of the proportions receiving a psychological assessment following a referral. However, Pearson's chi-square analysis revealed no statistically significant association between ethnicity and receiving psychological assessments following referral ($\chi^2$ (5, *n* = 599) = 7.98, *p* > 0.05).

The Black ethnic group, Asian ethnic group, Mixed ethnic group and Other ethnic group appeared to have higher than average rates of declining psychological assessments.

However, a Pearson's chi-square analysis revealed no statistically significant association between ethnicity and declining assessments ($\chi^2$ (5, $n = 599$) = 5.69, $p > 0.05$)

**Table 2.** Indicators of access and engagement with the burns clinical psychology service across ethnic groups.

| Ethnic Group (No. of Referrals) | No. of Patients Receiving Assessment (%) | No. of Patients Declining Assessment (%) | No. of Patients Not Attending Assessment (%) | No. of Patients Receiving Therapy (%) |
|---|---|---|---|---|
| White British (471) | 322 (68.4%) | 125 (26.5%) | 24 (5.1%) | 139 (30.0%) |
| Other White (38) | 25 (65.8%) | 8 (21.1%) | 5 (13.2%) | 11 (29.0%) |
| Black ethnic group (28) | 18 (64.3%) | 9 (32.1%) | 1 (3.6%) | 9 (32.1%) |
| Asian ethnic group (43) | 26 (60.5%) | 14 (32.6%) | 3 (7.0%) | 10 (23.3%) |
| Mixed ethnic origin (10) | 3 (30%) | 5 (50%) | 2 (20%) | 1 (10%) |
| Other ethnic group (9) | 5 (55.6%) | 4 (44.4%) | 0 (0%) | 0 (0%) |
| Total (mean) | 399 (66.6%) | 165 (27.5%) | 35 (5.8%) | 170 (28.4%) |

Furthermore, patients from the Other White, Asian ethnic and Mixed ethnic origin groups seemed to have higher non-attendance rates compared to the mean. However, a Fisher's exact test revealed no statistically significant association between ethnicity and non-attendance of assessments (two-tailed test, $p > 0.05$).

Patients in the Asian ethnic, Mixed ethnic origin and Other ethnic origin groups appeared to have lower than average rates of receiving psychological therapy. However, Pearson's chi-square analysis revealed no statistically significant association between ethnicity and receiving therapy ($\chi^2$ (5, $n = 599$) = 6.28, $p > 0.05$).

Finally, the median number of therapy sessions completed across all patients (excluding those in the Unknown group) was 3 (IQR = 6) sessions. Patients in the Asian ethnic group received the highest number of sessions (median = 6; IQR = 12.75), followed by the Black ethnic group (median = 5; IQR = 8), the Other White group (median = 3; IQR = 4) and the White British group (median = 2; IQR = 5.25). There was only one patient in the Mixed ethnic origin group who received sessions and they received four sessions. However, Kruskal–Wallis analysis revealed no statistically significant difference between the groups in terms of the number of psychological therapy sessions completed (H (4) = 4.16 $p > 0.05$).

## 4. Discussion

This is the first study to explore the ethnic diversity of referrals to a burns clinical psychology service and associations between ethnicity and indicators of access and engagement. Analysis revealed a statistically significant association between ethnicity and being referred to the burns clinical psychology service. Post hoc analyses showed that patients within the White British and Other ethnic groups were less likely to be referred to the burns clinical psychology service, whereas patients within the Black and Asian ethnic groups were more likely to be referred. It is possible that patients from Black and Asian ethnic backgrounds may present with higher psychological distress following a burn injury, which would explain the increased referrals. Indeed, some previous research has revealed a link between ethnic minority status and increased appearance concerns and lower social community reintegration following burn injuries [19,20]. The negative impact of racism on a range of mental health problems more widely has also been reported [11], and it is possible that increased referrals in these ethnic minority groups represent pre-existing mental health problems and/or a vulnerability to experiencing psychological difficulties post burn injury because of their ethnic minority status. It is also known that patients from ethnic minority backgrounds have difficulties accessing more general mental health or psychological therapy services due to racism and mistrust [3,4], which might also enhance their vulnerability to difficulties following burn injuries if accessing help for pre-existing difficulties has been difficult. Further research could benefit from extending the exploration of the relationship between ethnicity and referrals to burns clinical psychology services to include information about pre-existing mental health difficulties and presenting psychological difficulties related to the burn injury.

In relation to indicators of engagement with the service, analyses revealed no statistically significant associations between ethnicity and receiving, declining or not attending psychological assessments or receiving psychological therapy. Furthermore, there was no statistically significant difference in the number of psychological therapy sessions completed between the different ethnic groups. The findings of the current study therefore suggest that patients from ethnic minority backgrounds may not have experienced the same difficulties engaging with the burns clinical psychology service as has been found in previous studies that have considered barriers to engaging with general community psychology or mental health services [3–9]. This may be due to the burns clinical psychology service being embedded within the burns service at the affiliated hospital and the routine psychological screening that takes place for all patients admitted to the burns service, which may serve to reduce barriers to engagement. More research should be conducted investigating the experiences of ethnic minority groups in accessing and engaging with burns clinical psychology services. Qualitative methodologies may be particularly valuable in exploring patients' lived experiences of being referred to and (difficulties) engaging with services and the gathering of rich data. Burns clinical psychology services should regularly audit routinely collected data in relation to ethnicity. This is important in ensuring that services are accessible and acceptable to all patients and for services to consider improvements required. Despite no associations between ethnicity and indicators being found in this study, burns clinical psychology services should strive to continually seek the views of patients from ethnic minority groups and consider issues related to accessibility (e.g., through the provision of service leaflets in different languages and timely access to interpreters).

The current study also revealed that a significant proportion of ethnicity data collected by the burns clinical psychology service (14.3%) and affiliated hospital (21.6%) were classed as 'Unknown.' This is likely due to a systemic data collection issue within the organisation. This is in line with a finding from a recent report summarising a rapid evidence review related to ethnic inequalities in healthcare within the United Kingdom, which suggested that high-quality ethnicity data monitoring within healthcare services needs to be improved [3]. The report advised that mandatory guidelines should be put into place to ensure that patients' ethnicity is recorded, and recorded accurately, based on the finding that available research studies using clinical data often had substantial amounts of missing ethnicity data [3]. Clinically, this study suggests that burns services might benefit from evaluating whether they are recording patients' ethnicity routinely and ensuring that a system is in place to increase data collection where required. These systems should also be regularly evaluated so that future audits and research can be considered reliable and inclusive.

Limitations of the study include the significant proportion of data which had to be excluded from statistical analyses due to the patients' ethnicity being unknown, as detailed above. Furthermore, some of the ethnic groups had a low sample size.

Therefore, the findings should be interpreted with some caution. The study was conducted in a burns service in the UK and may not be generalisable to other countries or indeed other UK burns services that serve differing populations in regard to ethnic diversity.

## 5. Conclusions

In conclusion, patients from ethnic minority groups did not appear to have significant difficulties engaging with the burns clinical psychology service studied. However, further research has been recommended. Clinically, the study may point to the importance of clinical psychologists being embedded and visible within burns services to reduce barriers to engagement in ethnic minority groups. The study also highlights the importance of regularly auditing routinely collected data in relation to ethnicity to ensure that services are accessible and acceptable to all patients and consider service improvements. Burns clinical psychology services should also seek the experiences of patients from ethnic minority groups and consider issues related to accessibility. Finally, burns services should evaluate whether they are recording patients' ethnicity routinely and ensure a system is in place to collect this information.

**Author Contributions:** Conceptualization, L.S.; methodology, L.S. and L.B.; formal analysis, L.S., I.H. and L.B.; data curation, I.H. and L.B.; writing—original draft preparation, L.S., I.H. and L.B.; writing—review and editing, L.S., I.H. and L.B.; visualization, L.S., I.H. and L.B.; supervision, L.S.; project administration, L.S. All authors have read and agreed to the published version of the manuscript.

**Funding:** This research received no external funding.

**Institutional Review Board Statement:** The study was registered as a clinical audit by Nottingham University Hospitals NHS Trust, United Kingdom (ref. 22-332C).

**Informed Consent Statement:** Patient consent was waived due to the study being classed as a clinical audit using routinely collected clinical data.

**Data Availability Statement:** No new data were created for the purposes of this study. Only routinely collected clinical data were used and are therefore not available.

**Conflicts of Interest:** The authors declare no conflict of interest.

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
