# Peer review of "Associations between Ethnicity and Referrals, Access and Engagement in a UK Adult Burns Clinical Psychology Service"

_2673-1991, doi:10.3390/ebj4020017_

Round 1

Reviewer 1 Report

Thank you for the opportunity to review this paper on this interesting topic. I feel that the paper has potential but needs some improvements before it is suitable for publication.

My main concern are the analyses and the reference to meeting statistical assumptions. Please describe the statistical assumptions you mean in your data analysis paragraph.

I think the authors mean their relative small numbers in subgroups, however, they still use the Pearson's Chi Square test where the Fisher exact test is recommended for small numbers and I believe that subgroup analyses can be performed with the Fisher exact test, or at least more subgroups can be analysed than just two.

Another main point is the inclusion of individuals with unknown ethnicity. Why have the authors decided to include them? And why do they exclude them from the second analyses? This is inconsistent and makes it difficult to interpret your results. Consider to exclude unknown from your study.

Regarding the subgroup analyses and the two subgroups that have been made. It reads like Other White Ethnicity is included in the ethnic minority group instead of combined with White British, why is this done? And also unknown is included in the minority group. Unknown could include mainly White British.

In the results and discussion, some statements are made regarding the mean rates of therapy. However, no means are presented and should be added in a table. Consider creating a third table which includes the no of patients receiving therapy sessions (this column is confusing in Table 2) and add the means + SD.

In the results statements are made on outcomes being similar, higher, lower, etc, but no p-values are presented. Please add those.

The only p-value that is presented is in the last line, described as p>0.05. However, described in the discussion as being a trend. The reader needs the exact p-value to know whether to agree with you on that.

You conclude that ethinic minority groups are more likely to be referred to psych service. Do they also have more mental health complaints. Can you give any insights in that? And can you include that in your analyses. Besides, do you know the language of people and include that as a cofactor?

The authors acknowledge the importance of language and terminology. However, using the terms ethinicity and ethnic diversity without stating what they exactly mean. Please add the discription.

Are patients referred to psych services representable of the overall population on age and sex?

It is stated that results might not be generalisable to other countries. Are they to other UK burn units?

Please add the relevance for burn care in your discussion and conclusion. What can burn care learn from your results?

Please make the introduction more to the point. It is a very long introduction, especially for a brief report.

Author Response

Thank you for your helpful comments. We believe the manuscript has now been significantly strengthened as a result of these as well as additional comments from other reviewers. Please see the attachment. 

Reviewer 2 Report

Many thanks for the opportunity to review this interesting study. The manuscript is well-written and arguments well-articulated. Considering the numbers, I think it will be fair for the authors to include that the conclusions be interpreted with caution. Otherwise, the manuscript is in great shape already. 

Author Response

Thank you for your helpful comments. Please see the attachment. 

Reviewer 3 Report

Overall comments

This manuscript is well written and gives insight into an important area of health services that does not appear often in this field. The back ground section captures the key issues in the literature pertaining to ethnic inequalities in minoritised ethnic groups’ engagement and utilisation of health services. It was great to see the thoughtful linkage to the extant literature on ethnic inequities in psychological services. This greatly helped to set the scene for the rationale for the study.  

Major comment

I am well aware of the challenges in conducting research with minoritised ethnic group people especially when it comes to the small numbers in some ethnic group people. If the data allows, it is always important to disaggregate the ethnic group categories. However, in this instance, I’m concerned about the very small numbers in the minoritised ethnic group categories used in this study. The authors have rightfully pointed out this issue as a limitation. However, the numbers are still small and it leaves me wondering whether we can draw conclusions on the data. To increase numbers, perhaps the authors could use a longer follow up period instead of the 5 years to make it 10 years? Or perhaps  they can include children in their analyses? Would the authors consider aggregating some of the ethnic groups?  I understand they wanted to mirror the ethnic group categories used by Nottingham University Hospital NHS Trust but the numbers are so small within most ethnic groups.

Minor comments

In the discussion section, it would be great if the authors revisited some of the themes they introduced in the background sections regarding trust, racism and discrimination and use this lens to interpret the findings (if relevant)

Line 17-19: There is repetition of the word receiving

Line 58: Would authors reconsider the use of the term non-White. Perhaps something like minoritised ethnic groups?

Line 72: Typo on word Australia

Line 85: When referring to America in this sentence (and paragraph), is it North or South America?

Line 130-132: Given that the study includes adults, then it means that children were excluded, therefore, the last sentence may not be required.

Line 151: Related to the point above, what is the definition of adults given that the age range of included participants was 16-97yrs.

Author Response

(The authors gave the same response as above.)

Round 2

Reviewer 1 Report

Thank you for improving this paper. I feel like all my earlier comments have been addressed and recommend to publish this paper after including my last comment.

My only remaining comment is that using a non-parametric test (Kruskal-Wallis) and presenting means (SD) is not in line with each other. If data are not normally distributed, please present as median (IQR)

Reviewer 3 Report

Thank you for taking on board the comments and for revising the paper. Whilst it would be ideal to have larger numbers, I appreciate the difficulty of working with ethnicity data, especially given the fact that the paper focuses on a single burns unit. Thank you for addressing this in the discussion section. Apart from the minor points below, I have no further comments.

1.              Please reconsider use of the term ‘within ethnic groups’ in lines 168-171. Using the term means that one can interpret the section as meaning that there is a sub category from a larger pool of people from minoritised ethnic groups.  

2.              Typo: please correct the spelling of ‘proportion’ in line 168

3.              Typo: please delete the word ‘all’ from line 201

4.              Please reformat Table 2 to ensure that the rows align.

5.               Relatedly, Table 2 has a row for the totals but Table 1 doesn’t. For consistency, please consider adding a row for the total in Table 1.

6.               Paradies reference has been included twice ref 11 and ref 29. Please amend.
